# Extracellular Vesicles from Cerebrospinal Fluid of Leptomeningeal Metastasis Patients Deliver MiR-21 and Induce Methotrexate Resistance in Lung Cancer Cells

**DOI:** 10.3390/ijms25063124

**Published:** 2024-03-08

**Authors:** Ji Hye Im, Kyue-Yim Lee, Yoona Seo, Jiho Rhim, Yun-Sik Dho, Byong Chul Yoo, Jong Bae Park, Sang Hoon Shin, Heon Yoo, Jong Heon Kim, Ho-Shin Gwak

**Affiliations:** 1Department of Cancer Control, Graduate School of Cancer Science and Policy, National Cancer Center, Goyang 10408, Republic of Korea; 75262@ncc.re.kr (J.H.I.); 70564@ncc.re.kr (K.-Y.L.); 2Cancer Molecular Biology Branch, Research Institute, National Cancer Center, Goyang 10408, Republic of Korea; yoona.seo@ncc.re.kr (Y.S.); rhimj92@gmail.com (J.R.); 3Department of Cancer Biomedical Science, Graduate School of Cancer Science and Policy, National Cancer Center, Goyang 10408, Republic of Korea; yoo_akh@ncc.re.kr (B.C.Y.); jbp@ncc.re.kr (J.B.P.); heonyoo@ncc.re.kr (H.Y.); 4Neuro-Oncology Clinic, National Cancer Center, Goyang 10408, Republic of Korea; ysdho@ncc.re.kr (Y.-S.D.); nsshin@ncc.re.kr (S.H.S.)

**Keywords:** leptomeningeal metastasis, cerebrospinal fluid, methotrexate, extracellular vesicle, miR-21

## Abstract

Leptomeningeal metastasis (LM) is a common and fatal complication of advanced non-small cell lung cancer (NSCLC) caused by the spread of malignant cells to the leptomeninges and cerebrospinal fluid (CSF). While intra-CSF methotrexate (MTX) chemotherapy can improve prognosis, eventual MTX resistance deters continued chemotherapy. Recent studies have shown that increased miRNA-21 (miR-21) expression in the CSF of patients with LM after intraventricular MTX-chemotherapy is associated with poor overall survival; however, the molecular mechanisms underlying this resistance are poorly understood. Here, we confirm, in 36 patients with NSCLC-LM, that elevated miR-21 expression prior to treatment correlates with poor prognosis. MiR-21 overexpression or sponging results in a corresponding increase or decrease in MTX resistance, demonstrating that cellular miR-21 expression correlates with drug resistance. MiR-21-monitoring sensor and fluorescent extracellular vesicle (EV) staining revealed that EV-mediated delivery of miR-21 could modulate MTX resistance. Moreover, EVs isolated from the CSF of LM patients containing miR-21 could enhance the cell proliferation and MTX resistance of recipient cells. These results indicate that miR-21 can be transferred from cell-to-cell via EVs and potentially modulate MTX sensitivity, suggesting that miR-21 in CSF EVs may be a prognostic and therapeutic target for overcoming MTX resistance in patients with NSCLC-LM.

## 1. Introduction

Leptomeningeal metastasis (LM) is a condition in which cancer cells disseminate to the leptomeninges and disrupt cerebrospinal fluid (CSF) homeostasis, leading to severe neurological disability or death. Although LM can occur in up to 15% of all solid tumors, most originate from melanoma, breast, and lung cancers [1]. Patients with LM show very poor prognosis, with a median overall survival (OS) of 4 to 8 weeks without treatment [2,3]. While it is possible to treat patients with LM in a widespread manner throughout the neuraxis by intrathecal or intraventricular chemotherapy, improvements in overall patient survival are marginal and the benefits are largely palliative [4]. Furthermore, long-term administration of cytotoxic drugs for chemotherapy such as methotrexate (MTX), the most widely used antifolate agent for intra-CSF chemotherapy, is quickly complicated by drug resistance and is underscored by poor symptomatic or cytological response rates of approximately 10% [5]. In a recent study, screening of the CSF fraction after intraventricular MTX chemotherapy revealed that changes in the small non-coding RNA profile, particularly microRNAs (miRNAs), could also be associated with poor prognosis in patients with non-small cell lung cancer (NSCLC)-LM [6].

MiRNAs are small, regulatory, non-coding RNA molecules approximately 19–25 nucleotides in length that play key roles in the post-transcriptional control of gene expression [7,8]. A subset of these miRNAs known as “oncogenic miRNAs” (oncomiRs) have been identified as potential drivers of cancer development and metastasis [9,10]. Notably, miRNA-21 (miR-21), a well-studied oncomiR, is known to contribute to oncogenic physiologies [8,11]. In addition, miR-21 was found to be commonly overexpressed in various cancers and studies have found that the overexpression of miR-21 plays a significant role in tumorigenesis. In cancers such as glioblastoma, elevated miR-21 could facilitate cancer growth and prevent cell death; subsequent inhibition of miR-21 could also enhance drug sensitivity [12,13,14]. Likewise, in NSCLC, miR-21 was also found to affect cancer-associated phenotypes such as cell migration and chemosensitivity [15]. Interestingly, miR-21 was found to be highly abundant in extracellular vesicles (EVs) derived from the CSF of LM patients compared to those of healthy individuals, indicating that EVs and their miR-21 cargo may play a role in disease progression [6,16].

In our previous study, we observed, in a small number of patients, that miR-21 expression increased up to 18-fold in the CSF of an unfavorable prognostic group of patients with NSCLC-LM (*n* = 5) after intra-CSF MTX chemotherapy compared to pre-treatment levels, while the opposite was seen in the favorable prognostic group (*n* = 6) [6]. EVs are known to be vital components of cell-to-cell communication by delivering essential macromolecules such as proteins or functional RNAs to target cells, and can promote aggressive phenotypes in various cancers by modulating the local tumor environment [17]. Based on this, we hypothesized that CSF EVs containing high levels of miR-21 may induce MTX resistance via cell-to-cell communication in the CSF microenvironment.

Here, we explored whether (1) MTX resistance is correlated with high miR-21 expression in NSCLC cell lines and (2) pre-treatment CSF miR-21 expression correlates with poor prognosis in patients with NSCLC-LM who received intra-CSF MTX chemotherapy. Then, we verified that EVs from miR-21-enriched cell lines could transfer MTX resistance to MTX-susceptible cell lines with relatively low miR-21 expression, and the transfer of miR-21-enriched EVs from the CSF of patients with LM could induce MTX resistance via increased intracellular miR-21 expression in target cell lines. Our results suggest that EVs secreted through the CSF from patients with NSCLC-LM could induce MTX resistance via miR-21 transfer and that it is possible to target these miR-21-enriched EVs to overcome MTX resistance. Moreover, this study could provide useful information to clinicians, who should determine whether to stop or to continue intra-CSF MTX chemotherapy for patients with LM with dismal prognosis by measuring CSF EV miR-21 levels before the treatment and thereafter.

## 2. Results

### 2.1. Increased CSF miR-21 Levels in Patients with LM Were Associated with Poor Survival

LM is characterized as the metastasis of a tumor beyond its site of origin into the central nervous system through the meninges or CSF. The metastasized cancer cells grow in the subarachnoid space between the arachnoid and pia mater, and may continue to spread throughout the neuraxis (Figure 1A). Patients diagnosed with LM generally undergo intraventricular MTX chemotherapy (Figure 1B) to slow disease progression. We previously observed that NSCLC-originated LM (NSCLC-LM) patients who underwent MTX chemotherapy demonstrated abnormally elevated levels of extracellular miR-21 expression in the CSF [6].

To determine whether high miR-21 levels were unique among other miRNAs, the CSF miRNA profiles of NSCLC-LM patients (*n* = 7) were compared to those of healthy individuals (*n* = 3) via small RNA sequencing [16]. Among the 214 mature miRNAs that were examined, miR-21 was confirmed to be the most highly expressed miRNA (Figure 1C). Next, to evaluate whether CSF miR-21 levels were associated with the prognosis of NSCLC-LM patients who received intraventricular (ventriculolumbar perfusion) MTX chemotherapy (*n* = 36, Appendix A) [18], survival analysis was also performed. The OS among all patients was 191–395 days (median 237 days, 95% confidence interval 139–284 days, Appendix A), and the OS of patients with high CSF miR-21 levels (*n* = 9) was significantly shorter than that of patients with low CSF miR-21 levels (*n* = 27) (Figure 1D; log-rank test, hazard ratio 2.57, *p* = 0.00406). In addition, OS was negatively correlated with CSF miR-21 levels (*p* < 0.05, Pearson’s *r* = −0.152, Appendix A). The difference in the OSs of patients despite similar regiments of intraventricular MTX chemotherapy suggested a change in MTX sensitivity.

### 2.2. Cellular miR-21 Levels Are Inversely Correlated to MTX Sensitivity in NSCLC Cell Lines

To examine whether miR-21 expression could influence MTX sensitivity in NSCLC-LM, various NSCLC cell lines were selected for treatment with MTX to account for heterogeneity and cell line specificity. HCC-827, NCI-H226 (H226), A549, NCI-H1299 (H1299), NCI-H460 (H460), and HOP-62 cells were treated with MTX for 72 h and a sulforhodamine B (SRB) assay was performed to identify the half-maximal inhibitory concentration (IC_50_) of MTX in each of the six NSCLC cell lines. The IC_50_s of MTX for each cell line were 0.65, 0.20, 0.11, 0.08, 0.006, and 0.004 μM in HCC-827, H226, A549, H1299, H460, and HOP-62 cells, respectively (Figure 2A). In addition, to determine if the different levels of MTX sensitivity could be associated to the cellular miR-21 expression levels of these cells, real-time quantitative PCR (qRT-PCR) of the respective NSCLC samples was performed (Figure 2B). Interestingly, cellular miR-21 expression demonstrated strong negative correlation with MTX sensitivity in cells with higher miR-21 expression correlating with higher IC_50_ values (Figure 2C, Pearson’s *r* = 0.8527), thus indicating that miR-21 may have a negative effect on MTX sensitivity.

### 2.3. Modulation of miR-21 Expression Affects MTX Sensitivity and the PTEN-Akt Pathway in NSCLC Cell Lines

To verify whether the modulation of miR-21 expression could affect MTX sensitivity, low-miR-21-expressing H460 cells were engineered for the stable expression of primary miR-21 (pri-miR-21) through the transduction of a pri-miR-21 minigene (Figure 3A). The resulting overexpression of miR-21 in H460 was evaluated by qRT-PCR and showed a 14.8-fold increase in cellular miR-21-5p expression compared to the control (Appendix A). Furthermore, proliferation assays revealed that pri-miR-21-expressing H460 cells retained higher cell viability with increasing dosages of MTX compared to the control (Figure 3B). This suggested that increasing miR-21 expression in NSCLC cells could decrease MTX sensitivity.

Recent studies have explored the role of miR-21 in response to various drugs and suggested that miR-21 could promote chemoresistance in NSCLC through negative regulation of PTEN and PDCD4; consequently activating the Akt/Erk pathway [15,19]. Likewise, western blot analysis of miR-21-overexpressing H460 cells demonstrated similar changes with a decrease in PTEN and PDCD4 expression and an increase in phosphorylation of the Akt protein (Figure 3C), suggesting that miR-21 may be modulating MTX resistance via the PTEN-Akt pathway.

To confirm whether this phenomenon was reflected upon miR-21 depletion, high-miR-21-expressing A549 cells were transduced with an 11 × miR-21 binding site containing miRNA sponge (Figure 3D). Evaluation of cellular miR-21-5p expression in the miR-21-sponge-expressing A549 cells by qRT-PCR showed a 48% depletion compared to the control (Appendix A). Furthermore, cell proliferation decreased significantly at 0.01 μM MTX treatment in miR-21-depleted A549 cells, while a significant decrease in proliferation in the control group could be seen at a higher MTX dosage of 0.1 μM (Figure 3E). Thus, this indicated that depletion of miR-21 could increase MTX sensitivity in NSCLC cell lines. In addition, further analysis of these samples by western blotting revealed that miR-21 depletion increased the expression of PTEN and PDCD4; at the same time, decreased the phosphorylation of the Akt protein (Figure 3F). In sum, these data provide further evidence that miR-21 indeed modulates MTX sensitivity.

### 2.4. Extracellular Vesicles Transfer miR-21 into Recipient Cancer Cells

In previous observations, the abundance of miR-21 in the CSF of NSCLC-LM patients could be attributed to CSF-derived EVs [6,16]. These circulating EVs are well known to be able to deliver macromolecular cargo from cell-to-cell and contribute to cancer development and progression [20,21,22], implying that miR-21 could also be delivered by EVs.

To determine whether EVs could perpetuate miR-21 within the microenvironment, cell-to-cell transfer of EVs between NSCLC cell lines was examined first. EVs isolated from A549 cells were tagged with a PKH26 lipophilic dye and delivered to H460 cells. A total 16 h after treatment, the recipient H460 cells were fixed, stained with DAPI, and were subsequently assessed with confocal microscopy. PKH26 dots could be observed near the nucleus of the recipient cell lines indicating uptake of the A549-derived EVs (Figure 4A). For further validation of EV uptake in the H460 cells, PKH26 fluorescent signals from the recipient H460 cells were assessed via flow cytometry (Figure 4B), confirming that EV transfer could indeed occur between the two NSCLC cell lines.

Next, to evaluate whether the transferred EVs could deliver miR-21, a recently devised miR-21-controlled dual reporter-expressing system (luc-21-miRDREL) was used to detect changes in miR-21 levels within the EV recipients. In this system, an increase in miR-21 abundance results in the increased binding of miR-21 to its five repeating miR-21 targeting sites, which causes translation inhibition of firefly luciferase activity and can be normalized by *Renilla* luciferase activity (Figure 4C) [16,23,24,25]. H460 cell lines were transduced with the luc-21-miRDREL system to create a respective miR-21 sensor-bearing cell line. EV-recipient H460 sensor cells were then treated with 4.0 × 10^9^/mL EVs isolated from either control or pri-miR-21 minigene-expressing H460 cells for 18 h. Relative firefly luciferase activity was more significantly inhibited in the cells treated with the EVs derived from the miR-21 overexpressed cells compared to the control (Figure 4D, *p* < 0.05). Conversely, when the sensor cells were treated with EVs isolated from either control or miR-21-sponge-expressing A549 cells under the same conditions, the relative firefly luciferase activity increased 1.52-fold in the recipient H460 sensor cells treated with the EVs from the miR-21-depleted A549 cells compared to the control (Figure 4E).

### 2.5. EVs Containing miR-21 Induce MTX Resistance in Low-miR-21 Recipient Cancer Cells

To determine whether the delivery of miR-21 via EVs could affect cell proliferation and MTX sensitivity, cell proliferation was monitored after EV treatment using the IncuCyte real-time cell monitoring system. H460 cells were treated with either H460- or A549-derived EVs, in which H460-derived EVs showed lower levels of miR-21 compared to A549-derived EVs. Upon treatment with H460-derived EVs, proliferation increased by 78.6% compared to the addition of EV-depleted media (from 4.2-fold to 7.5-fold, *p* < 0.01), and treatment with higher-miR-21-expressing A549-derived EVs increased proliferation by 120% (Figure 5A, from 4.2-fold to 9.2-fold, *p* < 0.01).

To confirm that the increased cell proliferation was dependent on miR-21 delivered by EVs, H460 cells were treated with EVs derived from pri-miR-21 minigene-expressing H460 cells, EVs derived from control-expressing H460 cells, and EV-depleted media. miR-21-abundant EVs increased H460 cell proliferation by 73.8% compared to the EV-depleted media (from 4.2-fold to 7.3-fold, *p* < 0.01, Appendix A). In a similar context, H460 cells were treated with EVs derived from miR-21-sponge-expressing A549 and control-expressing A549 cells. While miR-21-depleted EVs appeared to have elevated proliferation compared to EV-depleted media, miR-21-depleted EVs reduced proliferation by 27% when compared to control EV H460 cells (from 9.2-fold to 7.2-fold, *p* < 0.05, Appendix A).

Next, to test whether the transfer of miR-21 via EVs could affect the MTX resistance of the recipient cells, EVs derived from pri-miR-21 minigene-expressing H460 cells, EVs derived from control-expressing H460 cells, or EV-depleted media were treated to recipient naïve H460 cells. The cells were subsequently treated with increasing dosages of MTX and showed that EVs containing high levels of miR-21 diminished the inhibitory effect of MTX on cell proliferation (Figure 5B, *p* < 0.01). H460 cells were also treated with EVs derived from miR-21-sponge-expressing A549 cells, EVs derived from control-expressing A549 cells, or EV-depleted media. However, cell proliferation was increased in both miR-21-sponge-containing EVs and control-containing EVs compared to EV-depleted media (Figure 5C). We suspect that this result was caused by the relatively weak miR-21 suppression provided by the miR-21 sponge (48%, see Figure 4E and Appendix A), while both the A549 control EVs and the A549 miR-21 sponge EVs transferred enough miR-21 to reach saturation in the recipient cells.

### 2.6. EVs from the CSF of Patients with LM Promote MTX Resistance in Recipient Cells

To verify whether this occurred with high-miR-21-containing patient-derived CSF EVs, EVs from the CSF of three patients confirmed to have high miR-21 expression and were prepared. The LM particle-size distribution was 129 ± 6.6 nm (Appendix A). We verified EV-associated surface markers on the CSF-derived EVs using MACSPlex and ExoView (Appendix A) and their constituent miR-21 levels by droplet digital PCR (ddPCR) (Appendix A). The patient-derived CSF EVs were then treated to recipient luc-21-miRDREL-expressing H460 sensor cells. Luciferase activity assays confirmed the delivery of miR-21 to the recipient cells from the patient-derived CSF EVs (Figure 6A) and proliferation assays demonstrated that the patient-derived CSF EVs did affect recipient-cell proliferation (Figure 6B). Interestingly, MTX treatment of varying concentrations of H460 cells which received the patient-derived CSF EVs all demonstrated a reduced growth-inhibitory effect of MTX (Figure 6C, *p* < 0.01). Western blot analysis also showed that all three patient-derived CSF EV treatments reduced PTEN and increased phospho-Akt protein expression in recipient H460 cells (Figure 6D). These data suggest that CSF-derived EVs from patients with NSCLC-LM could induce MTX resistance by cell-to-cell delivery of miR-21 and the subsequent modulation of the PTEN/Akt pathway. Furthermore, this may indicate that MTX-resistant cancer cells could perpetuate MTX resistance by the delivery of miR-21 via EVs, thus playing a role in the rapid development of drug resistance in LM patients (Figure 6E).

## 3. Discussion

### 3.1. miR-21 Mediates MTX Drug Resistance

MiRNAs have been studied extensively for their role as regulators of gene expression, potential diagnostic markers, and potential therapeutic targets, especially in cancer [10,26]. While chemotherapeutic drug resistance is multi-factorial and involves complicated pathways, changes in miRNA profiles have been documented in cancer cells after treatment with chemotherapeutic drugs [27]. Many studies have suggested that overexpression of miR-21 is a prognostic factor in NSCLC, and increased miR-21 expression after chemotherapy has been observed [19,26,28]. In addition, miR-21 is also implicated in the development of chemotherapy resistance, as it is either directly involved in the regulation of tumor suppressor genes such as PTEN, PDCD4, and SMAD7 or indirectly involved in apoptosis inhibition [8,29,30]. Previous observations have shown that the dysregulation of miR-21 in NSCLC cell lines affects resistance to drugs including cisplatin and 5-FU [11,31,32]. We previously reported up-regulation of miR-21 in CSF of patients with NSCLC-LM after intraventricular MTX treatment, and these patients had poorer prognoses compared with patients with no change or reductions of miR-21 expression. Here, we showed that pre-treatment miR-21 expression was correlated with the OS of patients with NSCLC-LM who underwent intraventricular MTX chemotherapy. Furthermore, miR-21 expression in NSCLC cell lines was proportional to MTX resistance. The target genes of miR-21 are involved in multiple molecular pathways related to metastasis and drug resistance, such as PTEN which inhibits Akt phosphorylation by acting inversely to phosphoinositide 3-kinase (PI3K) [8,15,30]. PTEN expression was decreased in miR-21-overexpressing cells, with evidence of being suppressed by miR-21. We found that miR-21 overexpression induced PTEN suppression with increased Akt phosphorylation. Conversely, miR-21 downregulation resulted in increased PTEN expression with suppression of Akt phosphorylation. These results suggest that miR-21 augmented MTX resistance partly via the PTEN-Akt pathway.

### 3.2. EVs Can Modulate the Phenotype of Recipient Cells including Chemo-Resistance

There is accumulating evidence that EVs facilitate intercellular communication by functioning as cargo containers for biomolecules, and that EVs can change the phenotype of cancer cells to facilitate metastasis [21,22,33]. Among the biomolecules carried by EVs, miRNAs have been found to play roles in tumor-induced inflammation, angiogenesis and immune modulation to facilitate tumor spreading, and tumor escape from immune surveillance [34]. Modulation of cellular miR-21 levels has been shown to cause changes in chemo-sensitivity [31], however direct evidence of EV-mediated modulation of chemo-sensitivity via miR-21 expression remains elusive, in contrast to the more widely available indirect evidence based on cancer cell proliferation rates [35]. Ren and colleagues verified that EVs from hypoxic mesenchymal stem cells promote NSCLC growth by delivering miR-21, which downregulates PTEN [36]. In another study, Vella and colleagues observed EV-mediated drug resistance in melanoma cell lines using a similar approach describe in the current study [20]. They transferred EVs from BRAF inhibitor-resistant melanoma to recipient melanoma cells, which resulted in dose-dependent activation of PI3K/Akt signaling and escape from BRAF inhibition. In their study, the EV-delivered biomolecule responsible for the drug resistance was not a miRNA but PDGFR-β. Another study by Cornell and colleagues demonstrated EV-mediated resistance to the CDK 4/6 inhibitor palbociclib in breast cancer cell lines [37]. They found that exosomal miR-432-5p mediated the transfer of resistance between neighboring cell populations via TGF-β pathway suppression. We provide further evidence of EV-mediated drug resistance in cancer by showing that EVs from cells with high miR-21 expression (A549 cells) increased MTX resistance in recipient cells with low miR-21 expression (H460 cells). Furthermore, delivery of EVs from cells after direct modulation of miR-21 expression resulted in increased (pri-miR-21) or decreased (miR-21 sponge) MTX resistance in recipient cells.

### 3.3. Clinical Implication of Acquired MTX Resistance in Intra-CSF Chemotherapy

Intra-CSF MTX has been the most frequently prescribed treatment for LM for decades because of its hydrophilic characteristics and low neurotoxicity [4,38]; however, its low response rate and marginal survival benefit suggest some inherent resistance in LM cancer cells [4]. Furthermore, we recently reported that intra-CSF MTX administration for LM treatment can cause cumulative toxicity in a setting of preceded brain radiation—more than 10 times that of intraventricular injection [39]. Thus, considering the fatal prognosis of LM despite various treatments, biomarkers representing treatment response in these terminal cancer patients with LM are absolutely required. Therefore, our observation that pre-treatment miR-21 levels in CSF were correlated with OS after MTX treatment is noteworthy. Based on our findings, these ‘high miR-21’ LM cancer cells can easily build up MTX resistance via EV shedding during treatment, and result in the least treatment benefit and the worst possible fatal toxicity. Our study could be used as a reference for clinicians to determine whether to continue intra-CSF MTX by measuring CSF pre-treatment and following miR-21 levels after treatment.

Although we used patient data from a prospective study of ventriculolumbar MTX perfusion chemotherapy in which all patients were expected to have the same treatment protocol with constant pharmacokinetics [18], it is difficult to suggest that miR-21 is the sole determinant of intra-CSF MTX chemotherapy response, and evidence of a molecular mechanism based on EV transfer in vivo via CSF is still preliminary. Clonal selection with the acquisition of genetic alterations is one hypothesis that would explain acquired chemo-resistance during repeated systemic chemotherapy. However, with intra-CSF chemotherapy, the transfer of EVs via CSF flow has not been confirmed at this level.

In the CSF environment, EVs from cancer cells spread via CSF flow to other cancer cells attached to leptomeningeal membranes. Thus, EVs may transfer MTX resistance during treatment. Although CSF can be collected repeatedly with minimally invasive techniques, such as a lumbar puncture or intraventricular reservoir puncture, after the development of LM, real-time molecular monitoring of EV transfer among cancer cells can hardly be achieved in patients with LM during intra-CSF MTX therapy. Researchers have focused on developing biomarkers in CSF, including miRNAs, metabolites, and EVs, but these need to be confirmed in large prospective studies. We verified that EVs from the CSF of patients with LM contained miR-21, and that transfer of LM CSF-derived EVs affected MTX resistance in NSCLC cells with low miR-21 expression. This suggests that EV-mediated miR-21 transfer between cancer cells can be one of the mechanisms of MTX resistance in patients with NSCLC-LM.

### 3.4. Limitations

We could not verify the transfer of miR-21 via EVs among LM cancer cells in vivo. The EV concentration necessary to induce phenotypic change (MTX resistance) was as high as 1.0 × 10^5^ EVs per mL of cell culture media. Although we previously observed that the EV concentration in the CSF of patients with LM ranged from 1.0 × 10^7^ to 1.0 × 10^8^ [6], it is not known whether that level of EVs, presumably containing various amounts of miR-21, is high enough to affect the phenotype of recipient cells. We verified that patients with LM and high pre-treatment miR-21 levels in their CSF had poorer prognoses. We also confirmed in vitro that cellular miR-21 levels were proportional to MTX resistance. EV transfer from NSCLC cell lines with high miR-21 expression to NSCLC cell lines with low miR-21 expression caused MTX resistance in the recipient cells. Furthermore, miR-21-containing EVs from the CSF of patients with NSCLC-LM were absorbed by NSCLC cells with low miR-21 expression and augmented MTX resistance in the recipient cells. Although we did not confirm EV transfer to neighboring LM cancer cells during intra-CSF chemotherapy, our results suggest that EVs released during/after chemotherapy may potentiate chemo-resistance in patients with LM. How miR-21 regulates MTX sensitivity remains unknown; however, we showed that EV-mediated miR-21 delivery into recipient cells modulated PTEN/Akt pathway activation. Our study suggests that miR-21 modulation may be a therapeutic method to overcome MTX resistance in patients with NSCLC-LM.

## 4. Materials and Methods

### 4.1. Clinical Samples and Patient Survival Analysis

All patients were diagnosed with LM based on CSF cytology and magnetic resonance imaging. CSF samples were collected after approval from the Institutional Review Board of the National Cancer Center, Republic of Korea (Identifier: 2014-0135), in accordance with the ethical guidelines outlined in the Declaration of Helsinki. Informed consent was obtained from all patients. All CSF samples were centrifuged for cell down (2000× *g* for 20 min) within 1 h, then stored at −80 °C until the next appropriate experiment. CSF miR-21 levels of 36 patients diagnosed with NSCLC-LM were assessed. Patients were divided into two groups according to their CSF miR-21 expression levels, and evaluated for survival analysis. The clinical characteristics of the patients including histologic subtypes and sampling site with measured miR-21 levels are summarized in Appendix A.

### 4.2. Cell Proliferation and Determination of the Half-Maximal Inhibitory Concentration (IC_50_) of MTX

Sulforhodamine B (SRB) and WST-1 assays were used to determine cell viability and proliferation which could extrapolate the half-maximal inhibitory concentration (IC_50_) of MTX [40]. Cell density was measured and 1 × 10^3^ cells/well were plated on a 96-well plate and incubated at 37 °C in a humidified atmosphere with 5% CO_2_ for 48 h. The cells were then treated with the indicated concentrations of methotrexate (MTX; JW Pharmaceutical, Seoul, Republic of Korea), and the absorbance was measured using a TECAN Infinite M200 PRO spectrophotometer (TECAN, Männedorf, Switzerland). Live-cell image analysis was used to visualize and measure the effect of CSF-derived EVs on the MTX sensitivity of recipient NSCLC cells. Cells were plated at a density of 1 × 10^3^ cells/well on a 96-well plate and treated with MTX for 72 h. Real-time live-cell confluence and images were obtained using the IncuCyte^®^ ZOOM system (Essen BioScience, Ann Arbor, MI, USA). Live-cell confluence was analyzed using IncuCyte^®^ ZOOM software (IncuCyte ZOOM 2016B, Essen BioScience).

### 4.3. EV Isolation and Quantification by Nanoparticle Tracking Analysis (NTA)

EVs were isolated from CSF or cell culture media by ultracentrifugation. EV isolation from patients’ CSF was restricted to that from the adenocarcinoma cell type. For EV separation from culture media, 2 × 10^6^ cells were plated on a 100 mm culture plate and incubated for 1 day. The plate was then supplemented with 10 mL fresh RPMI 1640 without fetal bovine serum (FBS) and incubated for 3 days. The media/CSF was then centrifuged at 2000× *g* for 20 min to eliminate cells and large debris, and the supernatant was centrifuged at 10,000× *g* for 20 min at 4 °C. Then, the supernatant was filtered through a 0.45 μm syringe filter (Thermo Fisher Scientific) and centrifuged at 116,000× *g* for 2 h at 4 °C in a disposable OptiSeal Polypropylene Tube (Beckman Coulter, Brea, CA, USA) with a NVT65 fixed angle rotor (Beckman Coulter) with maximal acceleration and deceleration. PBS was added to fill the OptiSeal tube completely. The EV pellets were used for downstream experiments immediately or stored at 4 °C for less than 1 week. EV-sized particles in the prepared EV sample were quantified using the NanoSight NS300 system (Malvern, Worcestershire, UK) and built-in software (ver. NTA3.2.16) as previously described [6]. Samples were diluted with PBS to an EV range of 10^6^ to 10^9^ and corrected to the input volume. All samples were measured with a sCMOS camera under identical detection conditions at room temperature and with a fixed camera level at 8 of 10 (Appendix A).

### 4.4. Luminescence-Based miR-21-Sensing System

A luminescence-based miR-21 sensor-bearing H460 cell line was established as previously described [16,23,24,25]. Briefly, after lentiviral infection, cells were treated with CSF EV concentrates and harvested 16–20 h later. The cells were lysed with Passive Lysis Buffer (Promega, Madison, WI, USA), and aliquots of lysates were analyzed by measuring luminescence signals with the Dual-Luciferase Reporter Assay System (Promega) in SpectraMax L (Molecular Devices, San Jose, CA, USA). The miR-21 sensor signal from firefly luciferase was normalized to that from *Renilla* luciferase. The normalized quantification data were used to compare relative luciferase activities [16,23,24,25].

### 4.5. Western Blot Analysis

Anti-PDCD4 (rabbit monoclonal, D29C6, #9535, Cell Signaling Technology, Danvers, MA, USA), anti-PTEN (rabbit monoclonal, 138G6, #9559, Cell Signaling Technology), anti-β-actin (ACTB, GTX629630, GeneTex, Irvine, CA, USA), anti-GAPDH (rabbit monoclonal, 14C10, #2118, Cell Signaling Technology), and anti-p-Akt (Ser473; rabbit monoclonal, #4060, Cell Signaling Technology) antibodies were used for overall western blot analysis. Whole-cell lysates were homogenized with RIPA [50 mM Tris-HCl pH 7.5, 150 mM NaCl, 1% sodium deoxycholate, 0.1% sodium dodecyl sulfate (SDS), 1% Triton™ X-100, 5 mM NaF, 2 mM sodium orthovanadate, 2 mM β-glycerophosphate, 2 mM EDTA, and protease inhibitor cocktail (Roche, Basel, Switzerland)] mixed with SDS gel-loading buffer, denatured for 5 min at 95 °C, and loaded onto Novex™ 4–20% Tris-Glycine Mini Gels (Thermo Fisher Scientific, Waltham, MA, USA). After electrophoresis, proteins were transferred to a PVDF membrane (Merck Millipore, Burlington, MA, USA). The membranes were blocked at room temperature with 5% nonfat dry milk in tris-buffered saline [50 mM Tris-HCI (pH 7.4) and 150 mM NaCl] with 0.1% TWEEN^®^ 20 (Amresco, Solon, OH, USA) and incubated overnight at 4 °C with primary antibodies. The membranes were then washed three times for 5 min with 1 × TBST prior to incubation with secondary antibodies. Horseradish peroxidase-conjugated anti-rabbit (Vector Laboratories, Newark, CA, USA) and anti-mouse IgGs (Vector Laboratories) were used as secondary antibodies. Enhanced chemiluminescence was detected and quantified by C-DIGIT (LI-COR, Lincoln, NE, USA) and/or X-ray film (AGFA, Mortsel, Belgium) exposure.

### 4.6. Statistical Analysis

Kaplan–Meier analysis was used to estimate cumulative survival, and the log-rank test was employed to compare patients’ OSs according to the pre-treatment miR-21 level using SPSS Statistics version 18 (SPSS Inc., Chicago, IL, USA). All experimental data are presented as the mean ± standard deviation determined from at least three independent experiments. Statistical significance was determined by a two-tailed Student’s *t*-test using R ver. 4.0.4 [CRAN, https://cran.r-project.org (accessed on 28 February 2024)] or Excel (Microsoft, Redmond, WA, USA). A *p*-value < 0.05 was considered significant, and the significance is described as * *p* < 0.05, ** *p* < 0.01, *** *p* < 0.001; ns, not significant on the figures.

## 5. Conclusions

This study provides comprehensive insight into the pivotal role of EV miR-21 from the CSF of LM patients in MTX resistance. We offer significant evidence that supports the targeting of EV miR-21 as a promising strategy for combating this aggressive form of cancer. Elevated miR-21 expression prior to treatment correlated with poor prognosis in 36 patients with NSCLC-LM. Furthermore, miR-21 overexpression or sponging results in a corresponding increase or decrease in MTX resistance, demonstrating that cellular miR-21 expression correlates with drug resistance. MiR-21 can be transferred from cell-to-cell via EVs and potentially modulate MTX sensitivity, suggesting that miR-21 in CSF EVs may be a prognostic biomarker as well as a therapeutic target for overcoming MTX resistance in patients with NSCLC-LM. Further research is required to validate these findings and develop effective targeted strategy for patients battling this devastating disease.

## Figures and Tables

**Figure 1 ijms-25-03124-f001:**
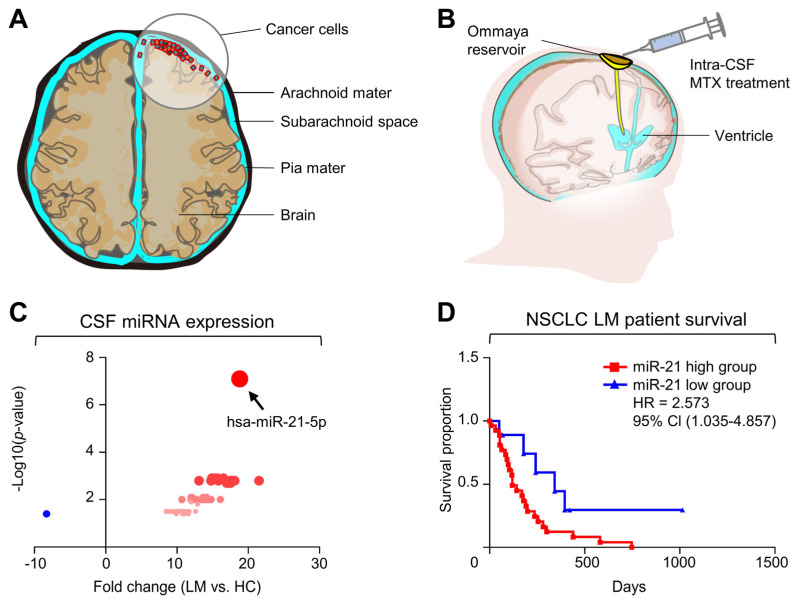
Increased CSF miR-21 correlates with poor overall survival (OS) in NSCLC-LM patients after intraventricular MTX chemotherapy. (**A**) Diagram showing LM. In LM, cancer cells grow into the subarachnoid space between the arachnoid and pia mater, which is filled with CSF. (**B**) Diagram of intra-CSF MTX administration. MTX can be injected into the ventricle by an Ommaya reservoir and spread through CSF flow. (**C**) Scatter plot depicting differences in expression of miRNAs in EVs from patients with NSCLC-LM and healthy controls (HC). (**D**) Kaplan–Meier plot of OS of patients with NSCLC-LM according to CSF miR-21 expression level. Log-rank test revealed significantly longer OS in the miR-21 low group compared with the miR-21 high group (*p* = 0.0406). HR, hazard ratio; CI, confidence interval.

**Figure 2 ijms-25-03124-f002:**
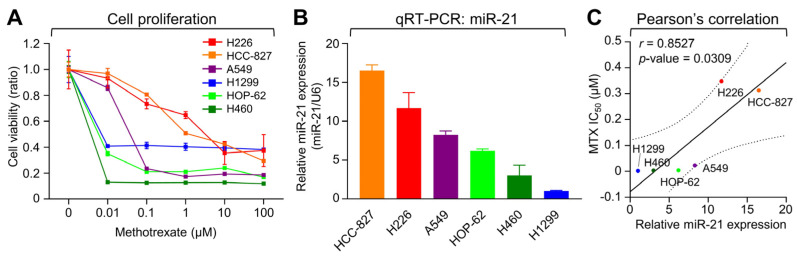
Cellular miR-21 expression shows a negative correlation with MTX sensitivity in NSCLC cell lines. (**A**) Inhibition of cell proliferation by MTX was quantified by a sulforhodamine B assay. Cells were treated with the indicated concentrations of MTX for 3 days. (**B**) Relative miR-21 expression in each cell line was measured by qRT-PCR and plotted after normalization to the U6 snRNA level. (**C**) Pearson’s correlation between the half-maximal inhibitory concentration (IC_50_) of MTX and relative miR-21 expression. Error bars represent ± standard deviation (*n* = 3). Statistical significance was verified by a two-tailed Student’s *t*-test.

**Figure 3 ijms-25-03124-f003:**
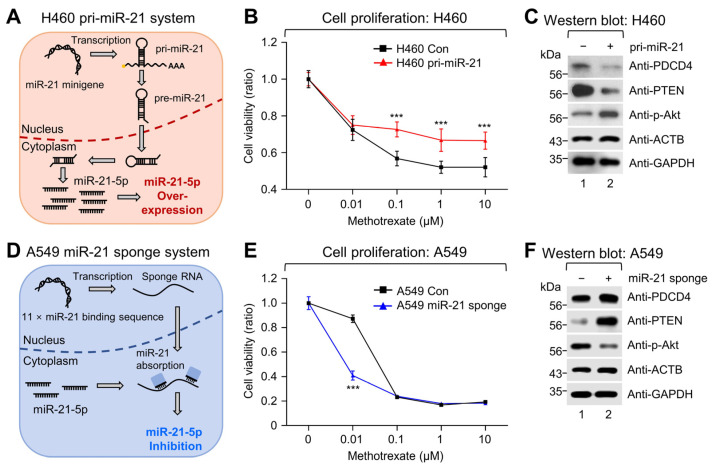
Modulation of miR-21 expression affects MTX sensitivity and activation of the PTEN-Akt pathway in NSCLC cell lines. (**A**) Diagram of the miR-21 overexpression strategy using the pri-miR-21 minigene-expressing lentiviral system. (**B**) Cell proliferation assays were performed using either pri-miR-21-expressing or control H460 cells to assess changes in proliferation after 72 h of MTX treatment via WST-1 assays. Changes in drug resistance-related downstream targets of miR-21 were assessed via western blot analysis of PTEN, PDCD4, and phosphorylated Akt expression in H460 (**C**) and A549 (**F**) cells. β-actin (ACTB) and GAPDH were used as loading controls. (**D**) Diagram of the miR-21 suppression strategy using the miR-21-sponge-expressing lentiviral system. (**E**) Effect of miR-21 depletion on cell proliferation was assessed by SRB assays of miR-21-sponge-expressing or control A549 cells after 72 h of MTX treatment. Error bars represent ± standard deviation (*n* = 3). Statistical significance was verified by a two-tailed Student’s *t*-test. *** *p* < 0.001.

**Figure 4 ijms-25-03124-f004:**
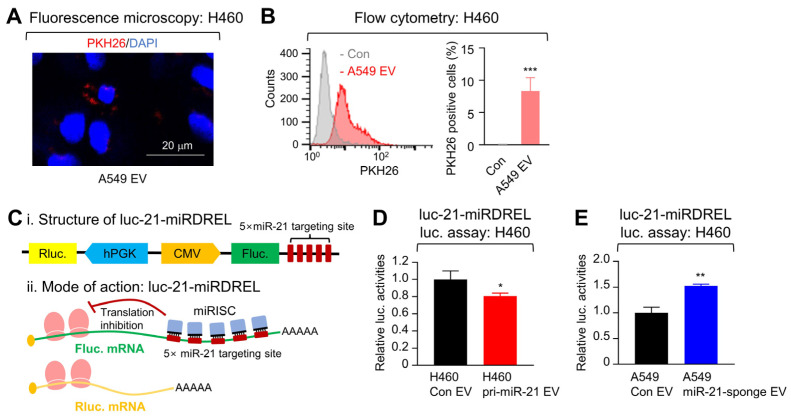
EVs can mediate cell-to-cell delivery of miR-21. (**A**) Fluorescence image of the internalization of A549-derived EVs isolated from the culture media stained with PKH26 dye and delivered to recipient H460 cells. The H460 cells were subsequently fixed with 4% paraformaldehyde solution and nuclei were stained with DAPI (4',6-diamidino-2-phenylindole). Images were obtained by confocal microscopy and a representative one is shown. (**B**) Flow cytometry of the aforementioned recipient H460 cell line confirmed and quantified the internalization of PKH26-stained A549-derived EVs. (**C**) Diagram of the miR-21 luciferase sensor system used to determine changes in cellular miR-21. Relative firefly luciferase activities were normalized by *Renilla* luciferase activities. (**D**) A luciferase assay of miR-21 sensor-expressing H460 cells treated with either H460-derived miR-21-overexpressed or control EVs was used to assess cell-to-cell delivery via EVs. (**E**) Similarly, miR-21 sensor-expressing H460 cells were also treated with either A549-derived miR-21 sponge or control EVs. Error bars represent ± standard deviation (*n* = 3). Statistical significance was verified by a two-tailed Student’s *t*-test, * *p* < 0.05, ** *p* < 0.01, *** *p* < 0.001.

**Figure 5 ijms-25-03124-f005:**
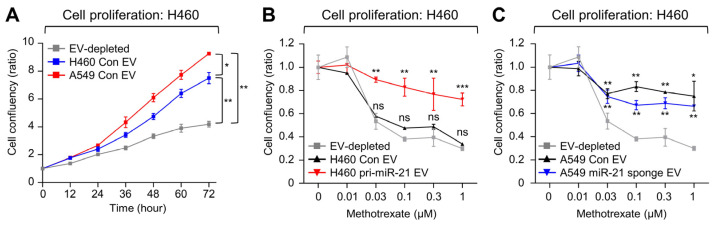
EVs containing miR-21 induce MTX resistance in recipient cells with low miR-21 expression. (**A**) The effect of EV-depleted medium, EVs from low-miR-21-expressing H460 cells, and EVs from high-miR-21-expressing A549 cells on the proliferation of recipient cells. Alterations in MTX sensitivity were observed after EVs from (**B**) pri-miR-21- and control-expressing H460 cells or (**C**) miR-21-sponge- and control-expressing A549 cells were treated to H460 recipient cells. Cell proliferations were monitored after treatment with the indicated concentrations of MTX for 72 h. Each set was also compared with EV-depleted media. Error bars represent ± standard deviation (*n* = 3), and the *p*-values compare each EV to the control. * *p* < 0.05, ** *p* < 0.01, *** *p* < 0.001, ns, not significant.

**Figure 6 ijms-25-03124-f006:**
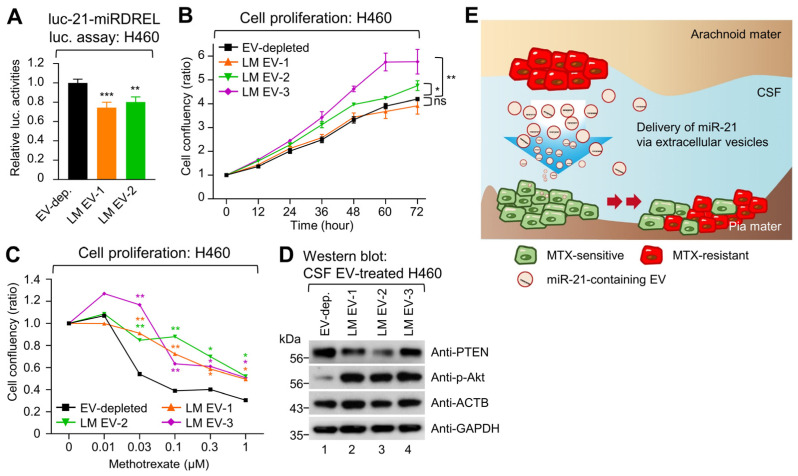
LM CSF-derived EVs promote MTX resistance in recipient cells with low miR-21 expression. (**A**) Luciferase assay of a miR-21-sensor-expressing H460 cells confirmed the delivery of CSF-derived miR-21-containing EVs into recipient cells. (**B**) The effects of the CSF-derived EVs on the proliferation of recipient cells with low miR-21 expression were analyzed. (**C**) Additional cell proliferation assays were performed to assess the effects of CSF-derived EVs on proliferation in recipient cells after 72 h of MTX treatment. Error bars in the graphs represent ± standard deviation and statistical significance was verified by a two-tailed Student’s *t*-test, * *p* < 0.05, ** *p* < 0.01, ***, *p* < 0.001, ns, not significant. (**D**) Changes in drug resistance-related downstream targets of miR-21 were assessed via western blot analysis of PTEN and phosphorylated Akt expression. ACTB and GAPDH were used as loading controls. (**E**) A schematic illustrating the propagation of MTX resistance to the MTX sensitive cells in LM through EV-mediated miR-21 delivery.

## Data Availability

The data that support the findings of this study are available on request from the corresponding author. The data are not publicly available due to privacy or ethical restrictions.

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
