# Peer review of "Extracellular Vesicles from Cerebrospinal Fluid of Leptomeningeal Metastasis Patients Deliver MiR-21 and Induce Methotrexate Resistance in Lung Cancer Cells"

_ijms, 2024, doi:10.3390/ijms25063124_

Round 1

Reviewer 1 Report

Comments and Suggestions for Authors

Review comment

The manuscript titled "Extracellular Vesicles from Cerebrospinal Fluid of Leptomeningeal Metastasis Patients Deliver MiR-21 and Induce Methotrexate Resistance in Lung Cancer Cells" presents a study on the role of extracellular vesicles (EVs) and miR-21 in mediating methotrexate resistance in lung cancer cells originating from leptomeningeal metastasis (LM). Authors pointed out that recent studies had shown that increased miRNA-21 (miR-21) expression in the CSF of patients with LM after intraventricular MTX-19 chemotherapy was associated with poor overall-survival. They concluded that miR-21 can be transferred from cell-to-cell via EV and potentially modulate MTX sensitivity, suggesting that miR-21 in CSF EV may be a prognostic and therapeutic target for overcoming MTX resistance in patients with NSCLC-LM. Even though the novelty is not quite apparent in this study, however, it still holds certain significance for clinical practice. Thus, mini-revision is recommended.

1.A deeper exploration of how the conclusions of this study could be translated into clinical practice or the development of new therapeutic approaches should be added in the section of discussion.

2.Authors should provide more details on the statistical methods used to analyze the data, ensuring clarity and reproducibility for readers and future researchers (Include more detailed explanations of the statistical tests used, ensuring that the choice of tests is appropriately justified based on the data type and study design).

3.Authors should elaborate on potential mechanisms by which miR-21 modulates drug resistance beyond the PTEN/Akt pathway and how these insights could lead to novel therapeutic interventions.

4.A more comprehensive discussion on the study's limitations is recommended, detailing any assumptions made during the analysis and exploring their potential impact on the broader applicability of the results.

Reviewer 2 Report

Comments and Suggestions for Authors

- The characterization of isolated exosomes requires the analysis of at least one exosomal marker, such as CD9 or HSP70, among others.

-Please highlight the novelty of the study at last paragraph of introduction.

Reviewer 3 Report

Comments and Suggestions for Authors

The research article “Extracellular Vesicles from Cerebrospinal Fluid of Leptomeningeal Metastasis Patients Deliver MiR-21 and Induce Methotrexate Resistance in Lung Cancer Cells” by Ji Hye Im et al. impressively reports experimental evidence that extracellular vesicle (EV)-mediated transfer of miR-21 affects cell proliferation and can modulate the chemo-sensitivity to methotrexate (MTX) of recipient cells. For the in vitro experiments, EVs from human lung cancer cell lines with modulated miR-21 levels and EVs containing miR-21 derived from the cerebrospinal fluid (CSF) of non-15 small cell lung cancer (NSCLC) patients with Leptomeningeal metastasis (LM) were used. The authors suggest that EVs transfer the drug resistance from cell to cell during chemotherapy via delivery of miR-21 and that miR-21 could be a prognostic factor for MTX chemoresistance and a therapeutic target in NSCLC LM patients.

Overall, the study was designed and conducted in a sensible way. The depicted data are convincing and accurate. Congratulations to the authors on the excellent work.

Nevertheless, I recommend that the authors address the following points:

1.    Figure 4b: Have the authors tried to increase the EV uptake efficiency? Only less than 10 % of the cells were PKH26 positive.

2.    Is it known which cell type in the CFS of LM patients delivers miRNA 21 for the cancer cells? Please add some statement in the discussion part.

3.    Did the authors use fresh patient-derived EVs or were the EVs unfrozen? Please mention this in the materials and methods section.

4.    Figure 6 A: Please add LM EV-3 luciferase activity if available.

Minor literal issues:

5.    Line 40: Typing error: reference “[1b]”. Please correct to “[1b]”.

6.    Lines 191 and 422: Replace “CO2” by “CO2 and change the bold text to normal text format.

7.    Passage 4.3: Please use unified line spacing.

8.    Figure 6a: Line 296: Redundant blank character after “While”.

9.    Line 364: Please remove paragraph after “In the LM”.

10.  Line 390-391: Please check the EV concentration. Do the authors mean 1.0 × 105, 1.0 × 107 and 1.0 × 108  EV per ml?
